# Modeling binary and graded cone cell fate patterning in the mouse retina

**Kiara C. Eldred**[1], **Cameron Avelis**[2], **Robert J. Johnston, Jr**[1]*, **Elijah Roberts**[2]*

**1** Department of Biology, Johns Hopkins University, Baltimore, Maryland, United States of America,
**2** Department of Biophysics, Johns Hopkins University, Baltimore, Maryland, United States of America

* robertjohnston@jhu.edu (RJJ); eroberts@jhu.edu (ER)

**Data Availability Statement:** Immunofluorescence images (https://osf.io/e5ckg/) and analysis code (https://osf.io/b438a) are available in the Open Science Framework database. All simulations were performed using a custom solver added to the

## Abstract

Nervous systems are incredibly diverse, with myriad neuronal subtypes defined by gene expression. How binary and graded fate characteristics are patterned across tissues is poorly understood. Expression of opsin photopigments in the cone photoreceptors of the mouse retina provides an excellent model to address this question. Individual cones express S-opsin only, M-opsin only, or both S-opsin and M-opsin. These cell populations are patterned along the dorsal-ventral axis, with greater M-opsin expression in the dorsal region and greater S-opsin expression in the ventral region. Thyroid hormone signaling plays a critical role in activating M-opsin and repressing S-opsin. Here, we developed an image analysis approach to identify individual cone cells and evaluate their opsin expression from immuno-fluorescence imaging tiles spanning roughly 6 mm along the D-V axis of the mouse retina. From analyzing the opsin expression of ~250,000 cells, we found that cones make a binary decision between S-opsin only and co-expression competent fates. Co-expression competent cells express graded levels of S- and M-opsins, depending nonlinearly on their position in the dorsal-ventral axis. M- and S-opsin expression display differential, inverse patterns. Using these single-cell data, we developed a quantitative, probabilistic model of cone cell decisions in the retinal tissue based on thyroid hormone signaling activity. The model recovers the probability distribution for cone fate patterning in the mouse retina and describes a minimal set of interactions that are necessary to reproduce the observed cell fates. Our study provides a paradigm describing how differential responses to regulatory inputs generate complex patterns of binary and graded cell fates.

## Author summary

The development of a cell in a mammalian tissue is governed by a complex regulatory network that responds to many input signals to give the cell a distinct identity, a process referred to as cell-fate specification. Some of these cell fates have binary on-or-off gene expression patterns, while others have graded gene expression that changes across the tissue. Differentiation of the photoreceptor cells that sense light in the mouse retina provides a good example of this process. Here, we explore how complex patterns of cell fates are specified in the mouse retina by building a computational model based on analysis of a

LMES software, which is available on our website:
https://www.robertslabjhu.info/home/software/lmes/.

**Funding:** RJJ and KE acknowledge support from the National Eye Institute R01EY025598 and the BrightFocus foundation G2019300. K.C.E. was a Howard Hughes Medical Institute Gilliam Fellow and was supported by the National Science Foundation Graduate Research Fellowship Program under grant 1746891. ER acknowledges support from the National Science Foundation under grant PHY-1707961. The funders had no role in study design, data collection and analysis, decision to publish, or preparation of the manuscript.

**Competing interests:** The authors have declared that no competing interests exist.

large number of photoreceptor cells from microscopy images of whole retinas. We use the data and the model to study what exactly it means for a cell to have a binary or graded cell fate and how these cell fates can be distinguished from each other. Our study shows how tens-of-thousands of individual photoreceptor cells can be patterned across a complex tissue by a regulatory network, creating a different outcome depending upon the received inputs.

## Introduction

How the numerous neuronal subtypes of the vertebrate nervous system are patterned is an ongoing puzzle in developmental neurobiology. Are neuronal subtypes distinct states generated by binary gene expression decisions? Or are they highly complex with ranges of graded gene expression? The answers likely lie somewhere in between, with some genes expressed in a simple switch-like fashion and other genes expressed across a range of levels to define cell fate. A challenge is to understand how cells interpret regulatory inputs to generate complex patterns of binary and graded cell fates across tissues. Here, we address this question in the context of cone photoreceptor patterning in the mouse retina.

Photoreceptors detect and translate light information into electrical signals, triggering the neuronal network yielding visual perception. There are two main classes of image-forming photoreceptors: rods and cones. Rods are mainly used in night vision, while cones are used in daytime and color vision. In most mammals, cones express S-opsin, which is sensitive to blue or UV-light, and M-opsin, which is sensitive to green light [1, 2].

The common laboratory mouse, *Mus musculus*, displays complex patterning of cone opsin expression across its retina, providing an excellent system to study binary and graded features of cell fate specification. The dorsal third of the retina is mostly comprised of cones that express M-opsin, and a minority that exclusively express S-opsin. In the central region, most cones co-express S- and M-opsin, with small subsets that express only S- or only M-opsin. The majority of the ventral region contains cones that co-express S- and M-opsin, with significantly higher levels of S-opsin compared to M-opsin [2–7]. Here, we expand upon these pioneering studies to examine cone patterning along the complete dorsal to ventral (D-V) axis of the mouse retina and quantitatively model how regulatory inputs influence cone cell patterning.

Cone subtype fate is not only characterized by opsin expression, but also connectivity. Two cone subtypes have been defined primarily on connectivity to downstream bipolar neurons. 3–5% of cones are "genuine" S cones that express S-opsin only and connect to blue-cone bipolars. The remaining cones express S-opsin only, M-opsin, or both S-opsin and M-opsin and do not connect to blue-cone bipolars [7]. The regulatory relationship between connectivity and opsin expression during cone subtype specification has not been established. In this work, we focus on the binary and graded nature of opsin expression, specifically examining this aspect of cone subtype fate.

Cone opsin expression is regulated by thyroid hormone (TH) signaling. TH and the nuclear thyroid hormone receptor Thrβ2 are important for activating M-opsin and repressing S-opsin expression [8]. TH exists in two main forms: T4, the circulating form, and T3, the form that binds with high affinity to nuclear receptors and acts locally to control gene expression [9, 10]. T3 levels are highest in the dorsal part of the mouse retina and decrease ventrally [8]. Deiodinase 2 (Dio2), an enzyme that converts T4 to T3, is expressed at high levels in the dorsal region

of the mouse retina, and is thought to maintain the gradient of T3 in the adult retina [11, 12]. T3 is sufficient to induce M-opsin expression and repress S-opsin expression [8].

Thrβ2, a receptor for TH, is expressed in all cones of the retina [13, 14]. Thrβ2 acts as a transcriptional repressor in the absence of T3 binding, and as a transcriptional activator when T3 is bound [15]. Thrβ2 activity is required for expression of M-opsin and repression of S-opsin [8, 16–20]. Additionally, RXRγ, a hetero-binding partner of Thrβ2, is necessary for repressing S-opsin in dorsal cones [13]. The transcription factors Vax2 and Coup-TFII, which regulate and respond to retinoic acid levels, have also been implicated in photoreceptor patterning [21, 22]. For this study, we focus on modeling the contributions of TH and Thrβ2 to cell fate outcomes.

We desired to quantitatively model cone fate specification in the mouse retina. Our current theoretical understanding of cell fate determination within a tissue describes individual cell types as distinct valleys on an "epigenetic landscape" [23–27]. Cells make fate decisions by transitioning to one of these "attractor" states on the landscape [28]. Differences in gene expression between the states give rise to phenotypic differences between cell types. However, clustering based on single-cell transcriptomics data alone misses subpopulations unless hidden variables are accounted for [29, 30].

Recently, computational work has also focused on developing mechanistic models of cell-fate decisions [31–33], especially the formation of patterns in time and space [34, 35]. Multi-scale approaches that combine probabilistic and deterministic models of tissues at the scale of individual cells have shown promise in helping to elucidate the details of tissue patterning [36–40]. The zebrafish and goldfish retina have been studied to model cell fate decision making based on anticlustering mechanisms that give rise to a lattice structure of differentiated cell types [41–43]. The highly variable arrangement of cone subtypes in the D-V axis of the mouse retina provides a paradigm to develop computational approaches that describe complex patterns of cell types across a tissue [3, 7, 44].

Here, we present a multiscale computational model describing the emergence of the complex arrangement of cone cells found in the adult mouse retina using both probabilistic and deterministic methods. We collect data for the model from analysis of immunofluorescence images of adult retina tissues to identify and map individual cones along the entire D-V axis of the mouse retina. Based on opsin expression in the individual cells, we find that terminally differentiated cones can be classified into two main subtypes: S-only cones and co-expression competent (CEC) cones. The S-only cones express S-opsin only, whereas the CEC cones express M- and/or S-opsins in opposing dorsal-ventral gradients, with higher levels of M-opsin in cones in the dorsal retina and higher levels of S-opsin in cones in the ventral retina. We then use the data to parameterize a mathematical model of a two-step cone patterning process. Step one is a binary choice between S-only fate and CEC fate. If CEC fate is selected, a second mechanism regulates S- and M-opsin expression in a reciprocal, graded manner, along the dorsal-ventral axis. Our quantitative modeling shows that the expression of S- and M-opsins in CEC cells are differentially activated based on dorsal-ventral patterning inputs from T3. Our model closely recapitulates cone patterning observed in the mouse retina and provides insights into how spatial patterning inputs regulate binary and graded features of cell fate in parallel.

## Results

### Characterization of cone subtype patterning in the mouse retina

To globally characterize patterning of opsin expression in the adult mouse retina at 2 to 8 months old, we first examined the relative intensity of S- and M-opsin expression in the D-V

and temporal-nasal (T-N) axes at low resolution for whole-mounted retinas. We immunos-tained and imaged S- and M-opsin proteins at 100X magnification (see Materials and Methods). Following image acquisition, we manually rotated each image so that the D-V axis was aligned vertically (Fig 1A, 1E and 1I). At this resolution, individual cells cannot be identified, so we instead subdivided each image using a 25 pixel x 25 pixel grid, which is an area containing approximately one to two cells. Within each bin of the grid, we counted the number of pixels that had significant S-opsin signal alone, M-opsin signal alone, or both M- and S-opsin signals. We then normalized each bin by the total number of pixels with expression in that bin. This calculation gave us the relative density of each photoreceptor type by location in the retina (Fig 1B, 1F and 1J).

Next, we quantified global differences in patterning in the D-V and temporal-to-nasal (T-N) dimensions. We averaged the binned density values to obtain the relative density as a function of either D-V (Fig 1C, 1G and 1K) or T-N position (Fig 1D, 1H and 1L). We observed distinct transitions in both S- and M-opsin expression along the D-V axis. High levels of M-opsin in the dorsal region exhibit a gradual transition to low levels in the ventral region (Fig 1C). In contrast, S-opsin shows a rapid transition from zero to high expression in the D-V axis (Fig 1G). As these opsins display an inverse yet non-complementary relationship, co-expression was most prominent in the middle third of the retina where these two transitions overlap (Fig 1K). We observed minimal variation in S- and M-opsin signal in the T-N axis (Fig 1D, 1H and 1L). We imaged and analyzed six wild-type retinas at this resolution and saw a similar pattern in each (S1 Table). Together, we observed differential graded patterning for S- and M-opsin expression along the D-V axis (Fig 1C, 1G and 1K).

## Analysis at single cell resolution reveals two distinct cone subtype populations

To further investigate photoreceptor patterning along the D-V axis, we next analyzed cone subtype specification at the single-cell level. For the same six retinas, we imaged S- and M-opsin expression at 200X magnification in a strip measuring approximately 600 μm x 6,000 μm aligned vertically along the D-V axis (Fig 2A). Previous studies analyzed ~500 μm in the dorsal-ventral axis centered on the transition region [7], whereas our approach enabled evaluation of the entire ~6,000 μm length of the retina.

At 200x magnification, we were able to distinguish and identify individual cells. We developed an analysis pipeline to identify the position, size, and boundaries of the outer segment of each cone cell, a process known as segmentation (see S1 Text). Overall, we identified ~250,000 total cells across six retinas. Using these outer segment boundaries, we calculated the expression intensity of M- and S-opsin for each cell. We classified cones into groups expressing M-opsin only (Fig 2E, 2K and 2Q), S-opsin only (Fig 2F, 2L and 2R), and S- and M-opsin co-expression (Fig 2G, 2M and 2S). The pipeline did not identify distinct morphological or size differences among the cone outer segments (S1 Fig). We found that the pipeline's accuracy and false positive rate were comparable to hand-scored retinas (see S1 Text).

After obtaining the outer segment boundaries of the cone cells, we quantified the density of cone subtypes based on opsin expression relative to D-V position. Consistent with our low-resolution analysis, we observed a gradual decrease in the abundance of cones expressing M-opsin in the dorsal to ventral direction (Fig 3A), contrasted by a sharp increase in S-opsin expressing cones (Fig 3B). We fit these curves to Hill functions to quantify the steepness of the transition (S2 Fig). The transition in the S-opsin expressing cells is extremely sharp with an average Hill coefficient of ~30 while the M-opsin transition is much more gradual with a coefficient of ~2–3.

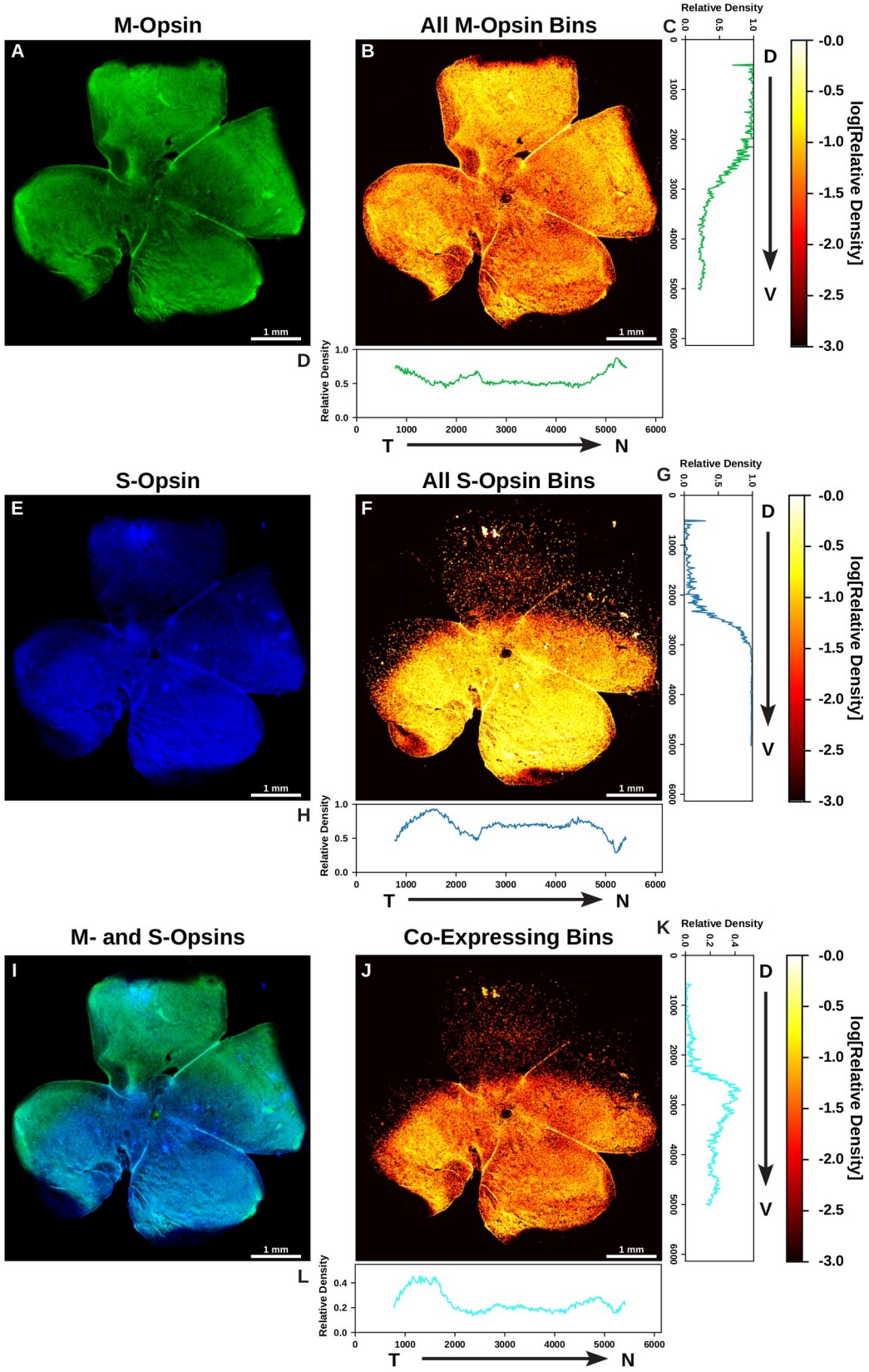

**Fig 1. Analysis of opsin expression intensity across the mouse retina. (A, E, I)** Whole mounted C57BL/6 mouse retina stained for M-opsin (green) and S-opsin (blue). **(B, F, J)** Heatmap displaying the log relative density of pixels that have opsin signal identified in a 25 mm² region. **(A)** M-opsin signal. **(B)** Heatmap of total M-opsin density bins. **(C)** Graph of the relative density of pixels that are expressing M-opsin summed horizontally (D—V). **(D)** Graph of the relative density of pixels that are expressing M-opsin summed vertically (T—N). **(E)** S-opsin signal. **(F)** Heatmap of total S-opsin density bins. **(G)** Graph of the relative density of pixels that are expressing S-opsin summed horizontally (D—V). **(H)** Graph of the relative density of pixels that are expressing S-opsin summed vertically (T—N). **(I)** M-opsin and S-opsin (co-expression) signal. **(J)** Heatmap of co-expressing opsin density bins. **(K)** Graph of the relative density of pixels that are co-expressing S- and M-opsin summed horizontally (D—V). **(L)** Graph of the relative density of pixels that are co-expressing S- and M-opsin summed vertically (T—N). T = Temporal, N = Nasal, D = Dorsal, V = Ventral.

To compare the transition region between retinas, we established a reference point to align the images. Since the S-opsin transition is sharp and an external reference is absent, we used the midpoint of the S-opsin transition from the fit as the reference point. We aligned all of the

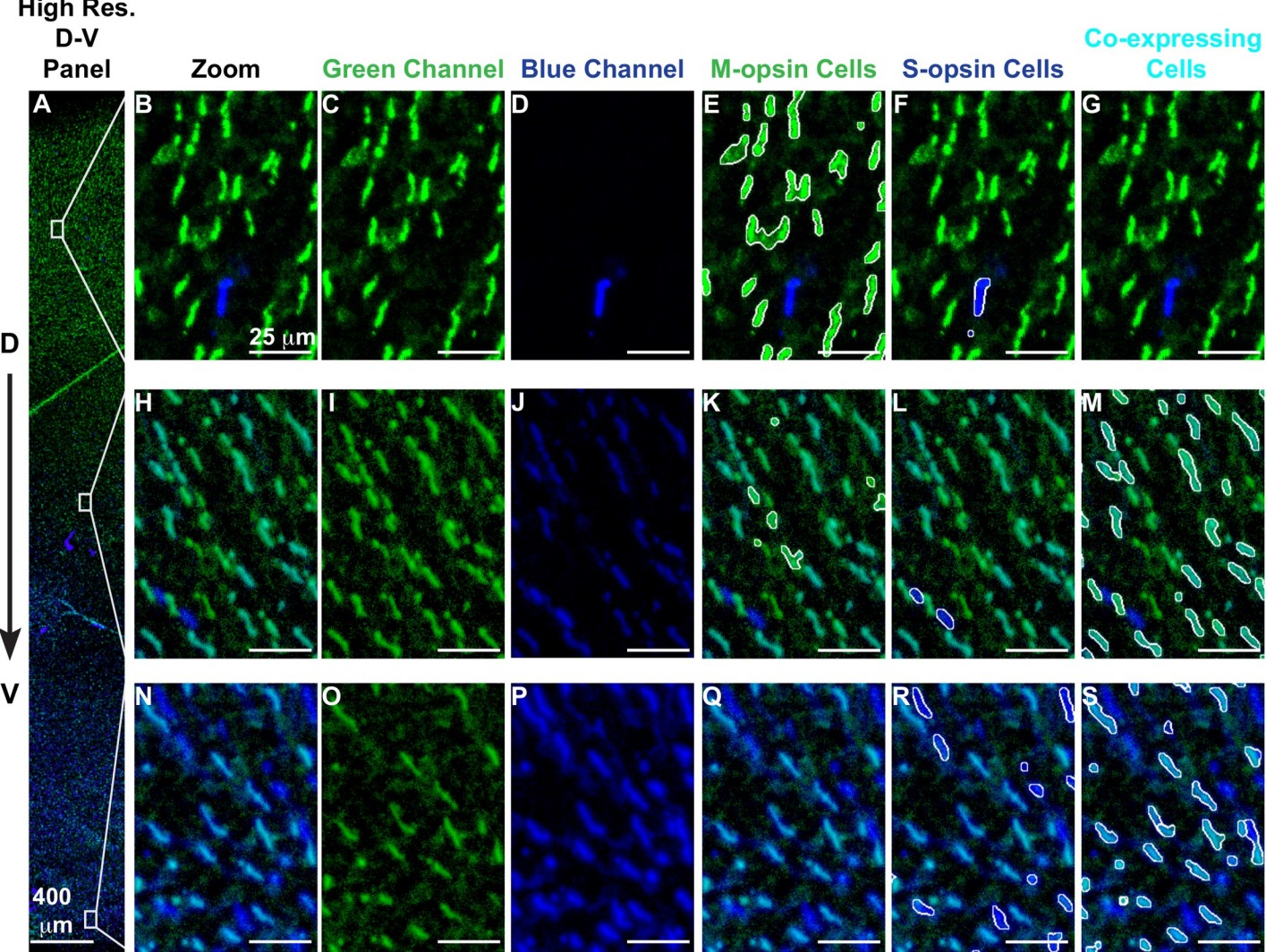

**Fig 2. Identification of cone subtypes. (A-S)** Retina stained with antibodies against M-opsin (green) and S-opsin (blue) **(A)** High-resolution region spanning the dorsal to ventral retina. **(B-G)** A region of the dorsal retina. **(F-M)** A region of the central retina. **(N-S)** A region of the ventral retina. **(B, H, N)** Blue and green channels. **(C, I, O)** Green channel only. **(D, J, P)** Blue channel only. **(E, K, Q)** White outline indicates identified M-opsin expressing cells. **(F, L, R)** White outline indicates identified S-opsin expressing cells. **(G, M, S)** White outline indicates identified co-expressing cells.

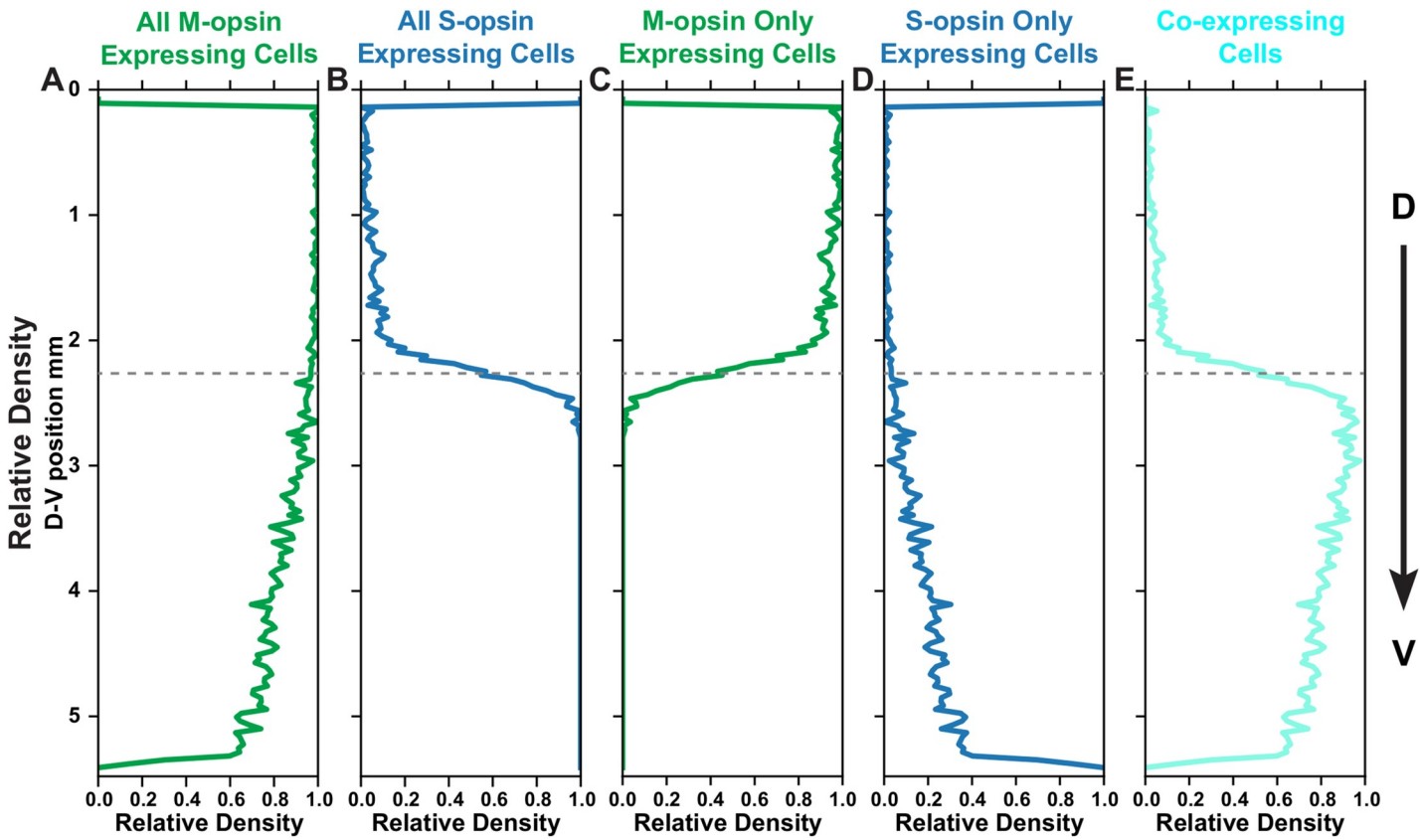

**Fig 3. Spatial distribution of M- and S-opsins in cone cells.** Relative density of a cone population summed horizontally across the image and displayed in the dorsal to ventral position. Dotted line represents midpoint of transition zone. **(A)** All M-opsin expressing cells. **(B)** All S-opsin expressing cells. **(C)** M-opsin only expressing cells. **(D)** S-opsin only expressing cells. **(E)** Co-expressing cells.

retinas and overlaid the transition fits (**S3 Fig**). The relative position of the S-opsin and M-opsin transitions are consistent from retina to retina, suggesting that the transitions in S-opsin and M-opsin expression are driven by a common effector. M-opsin only expressing cones decline at the transition point (**Fig 3C**), coincident with the dramatic increase in S-opsin expression (**Fig 3B**). At this transition point, cones begin to express both S- and M-opsins (**Fig 3E**) and the cone populations are very diverse, comprised of those expressing M-only, S-only, and varying levels of both S- and M- opsins (**Fig 2H–2M**). The fraction of S-only cells gradually increases from ~1% of cones in the dorsal region to ~20–30% in the ventral region (**Fig 3D and S4 Fig**). These analyses show the differential, inverse responses of S- and M-opsin expression to D-V patterning inputs on the individual cell level.

In a previous study, Haverkamp et al. measured differential opsin expression of cone cells in a window of ~500 μm near the transition point [7]. In agreement with our data, they observed that ~8–20% of cones expressed only S-opsin. Our results show that this measurement was part of a broader binary decision trend extending much further along the D-V axis in both directions. They also discovered that within this population, in the ventral region where S-only cones are more abundant, about 5% of S-opsin only cones contact S-cone bipolar cells and they classified these as genuine S-cones. These genuine S-cones are evenly distributed across the retina [7].

To distinguish classes of cone subtypes, we performed a cluster analysis. When considering all cones in the retina, there visually appear to be three groups of cell-types corresponding to

the three classifications that we defined earlier: S-only, M-only, and co-expressing. However, when we include the clustering analysis, a different pattern emerges (**Fig 4A**). Expression levels do not cluster around single points, but rather follow along manifolds in the high dimensional space. We used HDBSCAN, a density-based clustering analysis that connects regions of high local density, to generate clusters [45]. The method identified two distinct clusters of expression (**Fig 4B**). The two clusters are separated by a region of low density in the high dimensional space.

To study the properties of these clusters, we calculated the joint probability distribution for S- and M-opsin intensity in individual cells for each retina (**Fig 4C, S5 Fig**). First, we see an S-only cluster that has high and consistent expression of S-opsin while increasing in abundance along the D-V axis (**Fig 4D–4H, S6 Fig**). Interestingly, the other cluster changes position in a continuous way, gradually moving from low S-opsin expression and high M-opsin expression in the dorsal region, to moderate S-opsin expression and low M-opsin expression in the ventral region (**Fig 4D–4H, S6 Fig**).

Thus, these data suggest that the mouse retina contains two main subtypes of cones: 1. S-only cones that have high S-opsin expression independent of D-V position and 2. co-expression competent (CEC) cones that express S- and/or M- opsins dependent upon D-V position. In the ventral region there is a mixture of S-only cones and CEC cones that express M-opsin at a very low level. It is difficult to distinguish these two classes using only S-opsin expression, but as can be seen in **Fig 4**, the two populations are well separated when comparing both M- and S-opsin intensities. The S-only cones identified with our approach may, therefore, contain a subclass corresponding to the genuine S-cones identified by Haverkamp et al., but as we could not distinguish their connectivity to bipolar cells, we are only able to describe the populations of S-opsin only expressing cones.

## Expression levels of S- and M-opsin in cone cell subtypes

Having classified the major subtypes of cones and related their positions and opsin expression states, we next evaluated the D-V dependence of the opsin expression intensity in individual cones. We quantified opsin expression for all M-opsin expressing cones (**Fig 5A and 5F**), all S-opsin expressing cones (**Fig 5B and 5G**), M- and S-opsin CEC cones (**Fig 5C and 5H**), M-opsin only CEC cones (**Fig 5D and 5I**), and S-opsin only cones (**Fig 5E and 5J**) relative to their D-V positions within the retina.

In CEC cones, M-opsin expression levels decrease in the D-V axis, with the midpoint of expression level located at the transition point (**Fig 5A, 5C and 5D, S7 Fig**). In contrast, S-opsin expression in CEC cells is very low in the dorsal region and increases linearly in the D-V axis starting at the transition point (**Fig 5F, 5H and 5I, S7 Fig**). The slope of increase for S-opsin is steeper than for the M-opsin decrease (**S8 Fig**).

Compared to CEC cones, S-only cones have an overall higher expression level of S-opsin, particularly in the dorsal region (**Fig 5J** compared to **G** and **H**). M-opsin expression in S-only cones is significantly lower than the lowest M-opsin expression seen in CEC cones (**Fig 5E**). In **Fig 5B**, this difference can be seen as two distinct lines of density (**Fig 5B**, arrow heads). Together, these analyses defined the expression of S- and M-opsin in the two cone populations in relation to their D-V positions in the retina.

## Modeling cone subtype fate decisions

To interrogate how regulatory inputs could produce the complex pattern of binary and graded cell fates in the mouse retina, we developed a multiscale model describing the probability distributions of the cone subtype decisions (i.e. binary choice) and S- and M-opsin expression

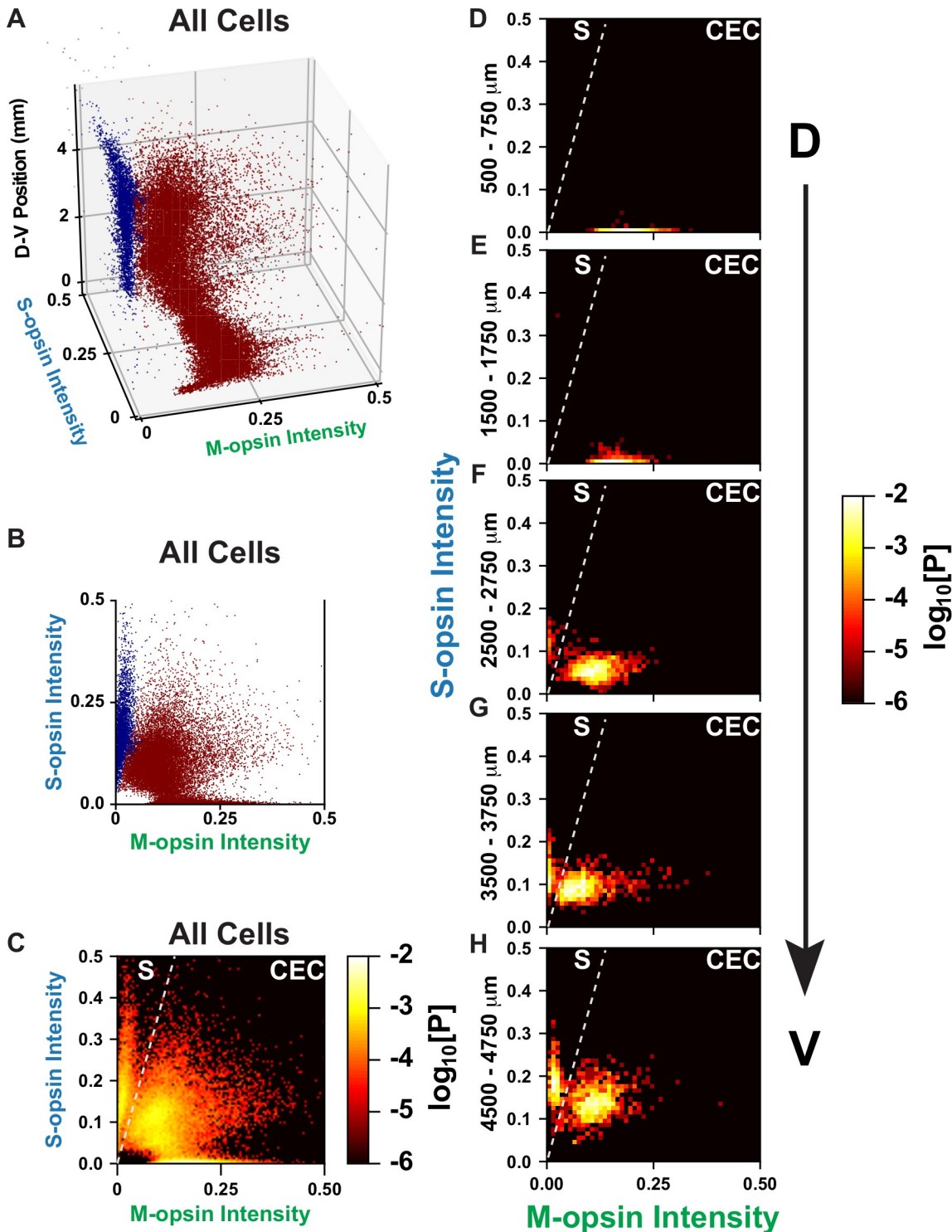

**Fig 4. S- and M-opsin intensities in cones.** (**A—B**) Clustering analysis of cone populations. Cluster one = dark blue; cluster two = maroon. (**C-H**) Cones are ranked according to the intensity of S- and M-opsin expression levels. Intensity values are represented in arbitrary units. Each point is colored according to the $\log_{10}$[Probability] of expression levels. A line is drawn on the graph to show the separation between the two discrete populations of S-opsin only and CEC cone populations. (**C**) All cones in the regions imaged. (**D**) Cones in the dorsal 500–750 mm. (**E**) Cones in the dorsal 1500–1750 mm. (**F**) Cones in the central 2500–2750 mm. (**G**) Cones in the ventral 3500–3750 mm. (**H**) Cones in the ventral 4500–4750 mm.

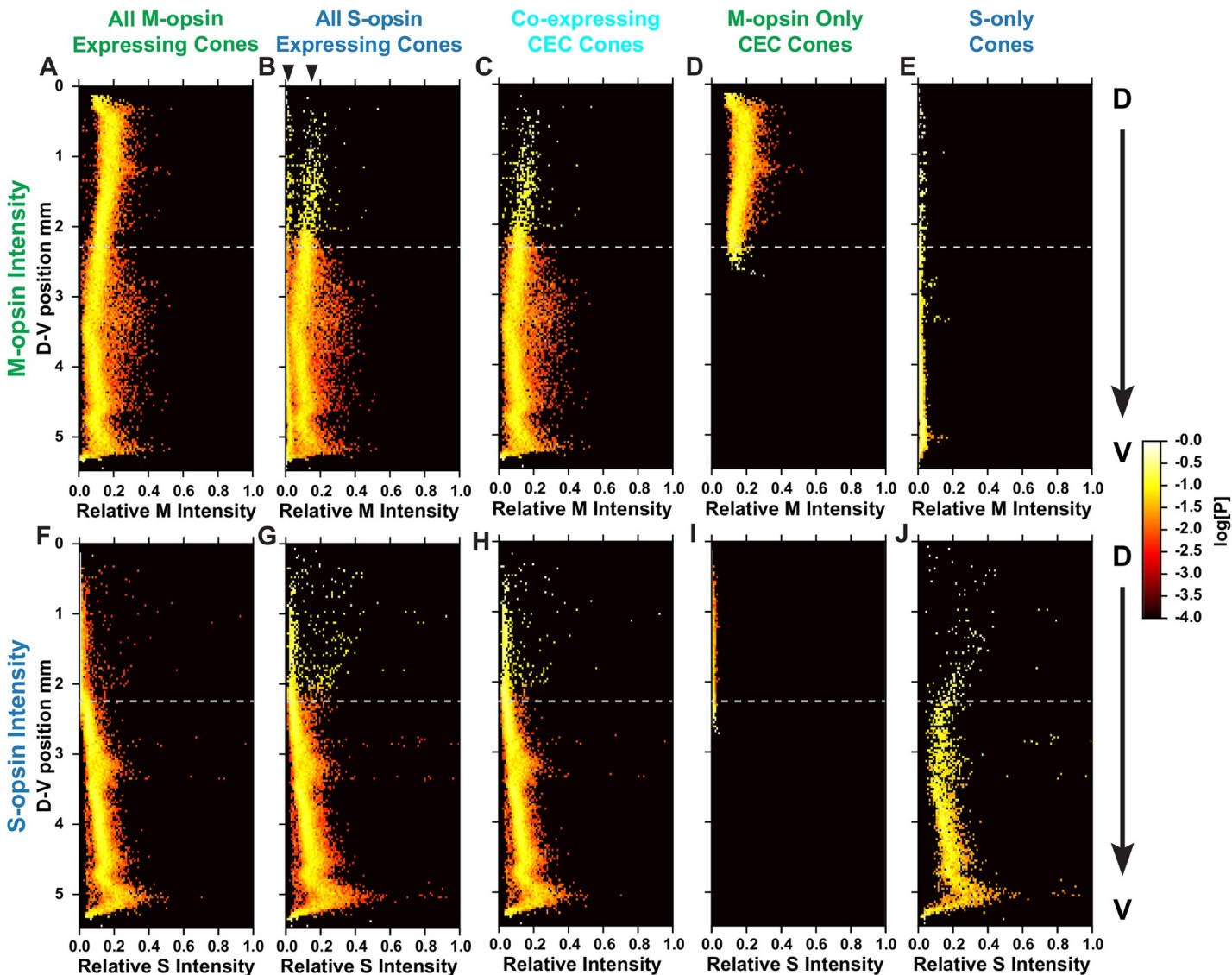

**Fig 5. Intensity of M- and S-opsins in cones.** Relative intensity of M- or S-opsin in a cone population (X-axis) is displayed as a function of dorsal to ventral position. Each point is colored according to the $\log_{10}$[Probability] of expression levels. **(A-E)** Relative intensity of M-opsin expression **(F-J)** Relative intensity of S-opsin expression **(A, F)** All M-opsin expressing cells. **(B, G)** All S-opsin expressing cells. For **(B)**, arrow heads mark two distinct groups of cells in the dorsal region. **(C, H)** CEC cones co-expressing both S- and M-opsins. **(D, I)** M-opsin only expressing CEC cones. **(E, J)** S-opsin only expressing CEC cones.

levels (i.e. graded) as functions of position along the D-V axis (**Fig 6,S1 Text**). We modeled a 5 mm x 1 mm x 5 μm section of the retina with the long dimension aligned with the D-V axis.

TH signaling activates M-opsin expression and represses S-opsin expression [8, 18]. T3 is a critical regulator of cone subtype fate in the human retina [17], and scRNA-seq data suggest that Thrβ2 is expressed in all mouse cones [46]. Though other diffusible factors and transcription factors play roles [13, 21, 22], TH signaling is the main and best-understood determinant of cone subtype fate. Thus, we built a simplified model of cone subtype specification based on the dorsal-ventral regulation of cone fates by the gradient of T3.

Within the modeled volume, T3 molecules diffuse according to the deterministic diffusion equation with constant concentration boundaries, establishing a D-V gradient. Roberts et al. (2006) reports a differential gradient in [T3] and [T4] between the dorsal and ventral regions

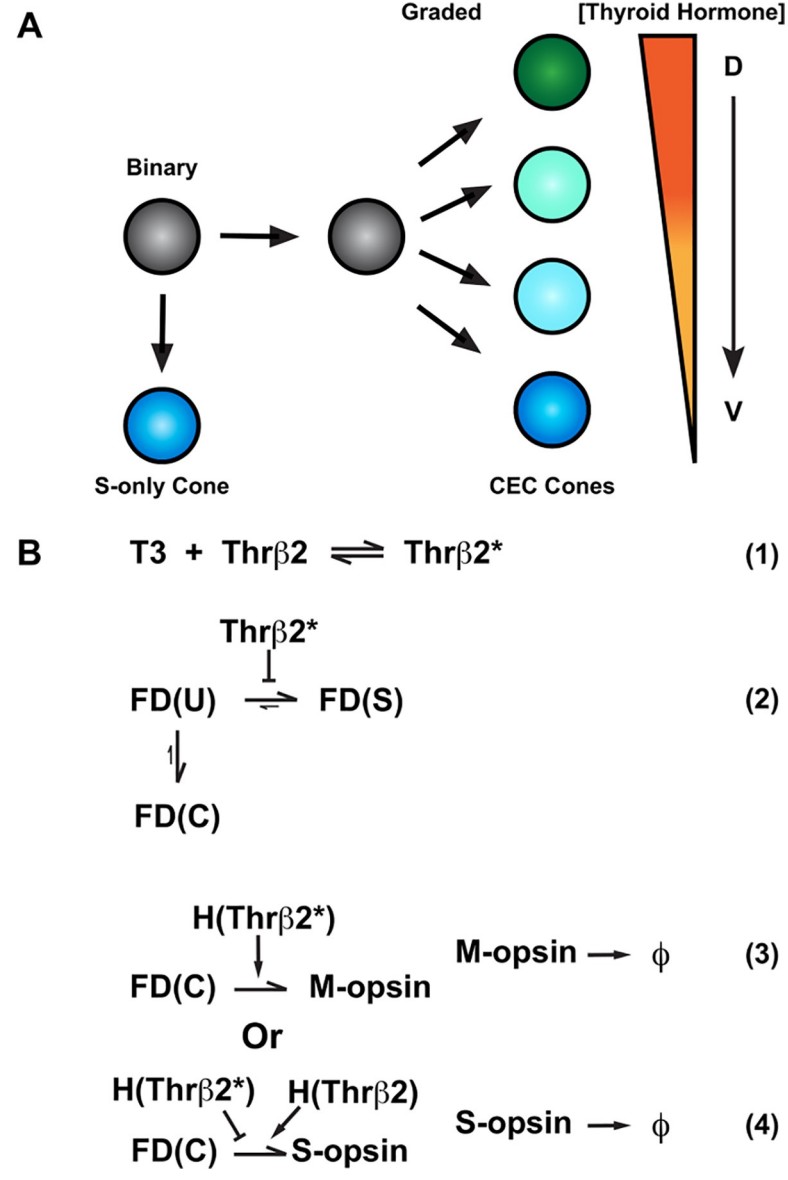

**Fig 6. Model for cone cell fate specification (A)** A naïve cell (grey) makes a binary decision between S-opsin only (blue) or co-expressing competent (CEC) cone fate (green, cyan or blue). The CEC cone expresses graded levels of M- and S-opsin dependent on the D-V concentration of thyroid hormone. **(B1-4)** T3 (Thyroid hormone), Thrβ2* (active Thrβ2 binding T3), FD (fate determinate function), U (undifferentiated cell), S (S-only cone), C (Co-expressing cone), H (Hill function), ϕ (degradation constant of opsin proteins). **(B1)** Binding of T3 to Thrβ2 activates Thrβ2 (Thrβ2*) **(B2)** Thrβ2 controls the binary decision between S-opsin only/FD(S) or CEC/FD(C) cone fate **(B3)** Thrβ2* promotes M-opsin expression **(B4)** Thrβ2* inhibits S-opsin expression, whereas inactive Thrβ2 promotes S-opsin expression.

of whole retina samples, however, it is not known what the intracellular concentrations of T3 are specifically in photoreceptor cells at a single cell level [8]. For this reason, we use relative values for [T3] to build a deterministic diffusion equation (**Eq. S1 in S1 Text**). Also, within the volume, we modeled ~23,000 individual cones spaced on a hexagonal grid (**Fig 7**). These cells randomly exchange T3 molecules with the surrounding deterministic microenvironment.

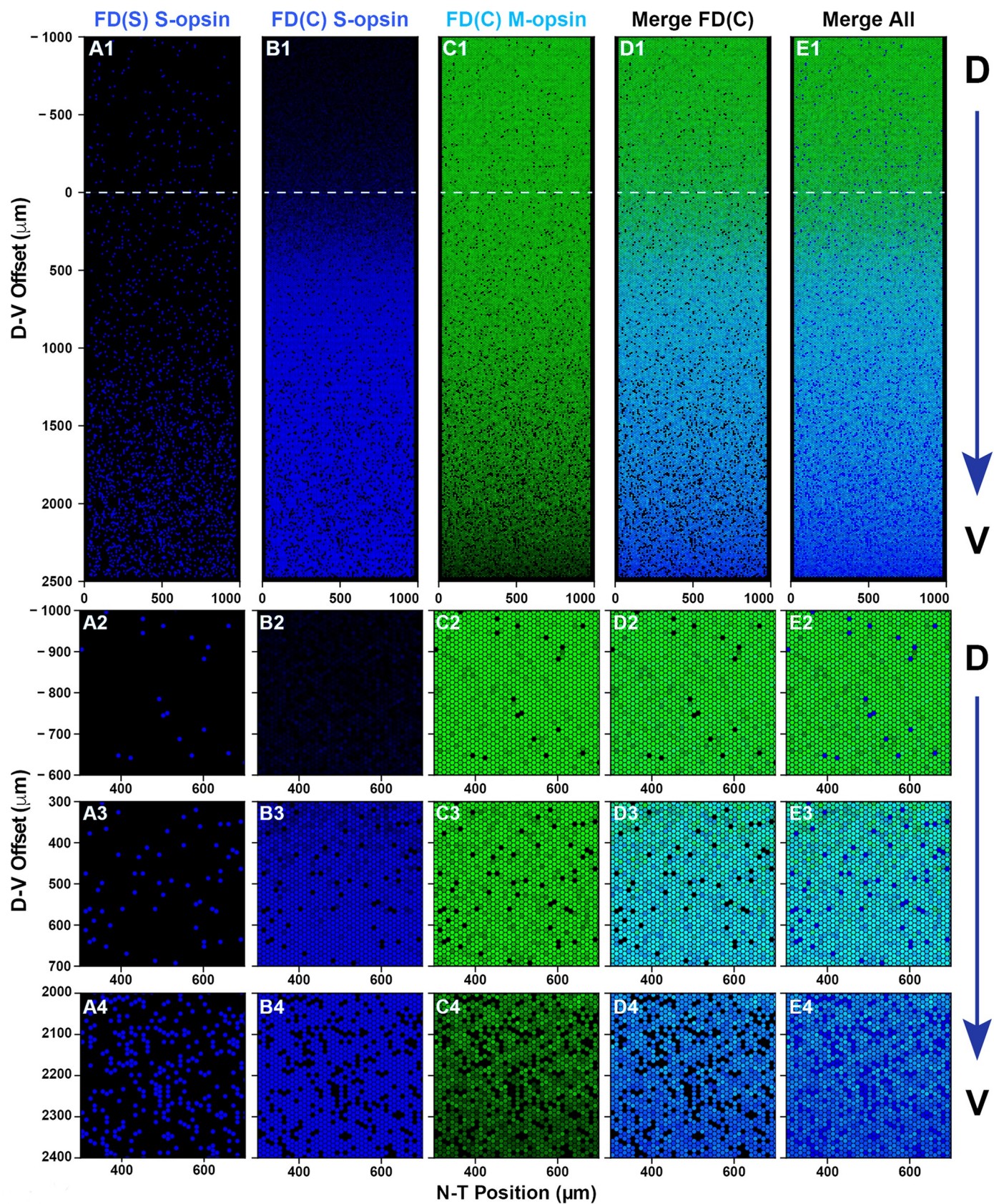

**Fig 7. Simulated cone mosaic produced by the quantitative model.** Simulated cone photoreceptor mosaic generated by the quantitative model displaying expression of S-opsin (blue), and M-opsin (green). A dorsal to ventral region is shown. **(A1, B1, C1, D1)** Complete simulated D-V strip. **(A2, B2, C2, D2)** Zoom in the dorsal region. **(A3, B3, C3, D3)** Zoom in the central region. **(A3, B3, C3, D3)** Zoom in the ventral region. **(A1-4)** S-opsin only cones. **(B1-4)** S-opsin expression in CEC cones. **(C1-4)** M-opsin expression in CEC cones. **(D1-4)** S- and M-opsin expression in CEC cones **(E1-4)** All cones including S-opsin only and CEC cones.

Within each cone, T3 can bind to and activate Thrβ2 (Thrβ2*), controlling both fate specification and opsin expression (**Fig 6B1-4**).

For the binary fate decision, we defined a fate determinant function, FD(X). Photoreceptors start in an undifferentiated fate, FD(U), and progress to either the FD(S) (S-only) fate or FD (C) (CEC) fate (**Fig 6B2**). Selection of the FD(S) fate is negatively influenced by Thrβ2* (**Fig 6B2**). Once cells enter the FD(S) fate, S-opsin is constitutively expressed at a high level regardless of D-V position (**Fig 6B2**). In FD(C) cones, M-opsin expression is induced by Thrβ2* (**Fig 6B3**). Conversely, S-opsin in FD(C) cells is negatively regulated by Thrβ2* and positively regulated by inactive Thrβ2 receptors (**Fig 6B4**). Full details of the model are given in the SI Methods 1.2, along with parameterization details.

## Model-based simulations recapitulate experimental cone patterning

To compare the output of our probabilistic model to experimental data sets, we ran a set of 100 individual simulations and calculated the probability distributions of various observables. **Fig 7** shows the output of one simulation. Moving from dorsal to ventral, the model reproduces the gradual increase in the fraction of S-only cones, FD(S) (**Fig 7A1-4**), as well as the sharp transition in CEC cones, FD(C), expressing S-opsin at the transition zone (**Fig 7B1-4**). Similarly, we observed the gradual decrease in the fraction of CEC cones expressing M-opsin (**Fig 7C1-4**). In the overlapping region, there are a significant number of cones that co-express both S- and M-opsins (**Fig 7D1-4**). In the dorsal region, a small number of S-only cones that highly express S-opsin are readily apparent (**Fig 7A1-2, 7E1-2**).

To characterize how well our model recapitulated the observed experimental cell distributions, we calculated the mean density of cells of various phenotypes as a function of D-V position (**S9 Fig**). These average density profiles compare well to the experimental density profiles shown in **Fig 3**. Together, cone fate patterning and expression levels are highly similar in our model and the imaged retinas: S-only cells (**Fig 7A2-4** compared to **Fig 2F, 2L and 2R**), S-opsin expression in CEC cones (**Fig 7B2-4** compared to **Fig 2E, 2G, 2K, 2M, 2Q and 2S**), and M-opsin expression in CEC cones (**Fig 7C2-4** compared to **Fig 2E, 2G, 2K, 2M, 2Q and 2S**).

We parameterized our model using the mean of all the retinas sampled, which exhibited retina-to-retina variability (**S2 Fig**, **S4 Fig**, **S8 Fig**). Therefore, it is not expected that our model will exactly recapitulate the patterning of any individual retina.

We next calculated the probability distributions of S- and M-opsin expression along the D-V axis for our simulation data (**S10 Fig**). The mean intensity of M-opsin in CEC cones gradually decreases as D-V position increases. The S-opsin distribution shows high expression in the ventral-most region, but has two separate populations in the dorsal region: the highly expressing S-only cells and the lowly expressing CEC cells. The CEC cones converge to zero S-opsin expression in the dorsal region while the S-only cones maintain high expression as they decrease in abundance. Because our simulated distributions are constructed from 100 independent simulations, the probability density of the S-only cones is much smoother than in the experimental data (compare **S7 Fig** and **S10 Fig**). The simulated expression features are in agreement with the experimental expression profile (**Fig 5**, **S7 Fig**).

We next related the joint probability distributions for the experimental (**Fig 8A–8E**) and simulated (**Fig 8F–8J**) data along the D-V axis. The simulated and experimental data show two

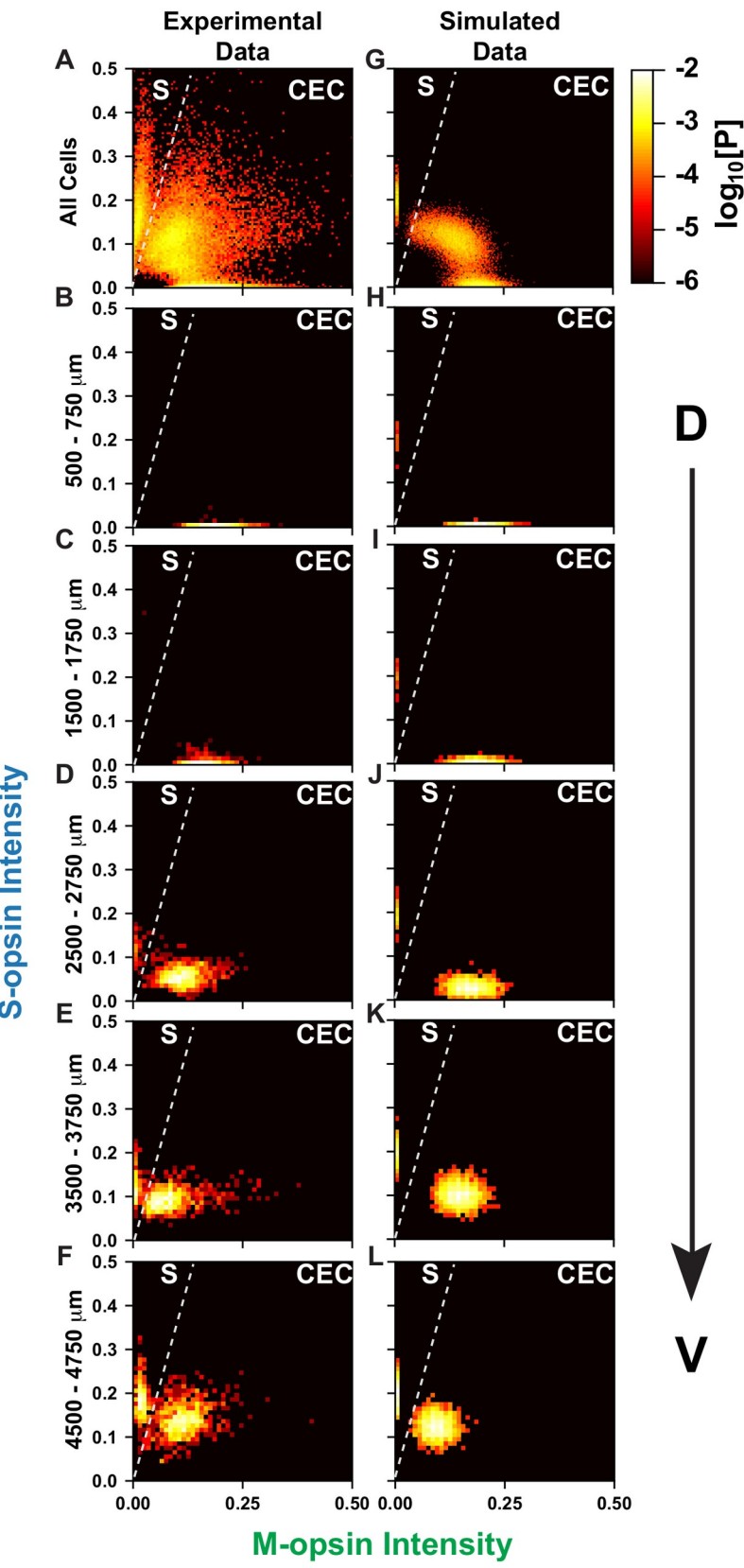

**Fig 8. D-V cone pattering in simulated and experimental data.** Cones are ranked according to the intensity of S- and M-opsin expression levels. Intensity values are represented in arbitrary units. Each point is colored according to the $\log_{10}$[Probability] of expression levels. **(A-F)** Experimental data, as seen in **Fig 4C–4H**. **(G-L)** Simulated data. **(A, G)** All cone cells **(B, H)** Cones in the dorsal 500–750 mm. **(C, I)** Cones in the dorsal 1500–1750 mm. **(D, J)** Cones in the central 2500–2750 mm. **(E, K)** Cones in the ventral 3500–3750 mm. **(F, L)** Cones in the ventral 4500–4750 mm.

distinct populations: 1) S-only cones with high S-opsin expression and no M-opsin expression whose expression levels are independent of D-V position, and 2) CEC cones that gradually change from high M-opsin and low S-opsin expression to moderate M-opsin and high S-opsin expression along the D-V axis. **S11 Fig** shows the joint probability distribution between S- and M-opsin expression within 250 μm D-V bins for all 100 simulations. In the high resolution simulated data, it is evident that the position of the CEC cell cluster gradually changes with D-V position. Our model closely simulates the experimental data, and supports the hypothesis that cells respond differentially to the same morphogen gradient, producing both binary and graded cell fates.

## Correlation between S-opsin and CEC fate decisions

Our quantitative simulations give us the capacity to test various hypotheses about retinal patterning. We wanted to know whether the gradual decrease in the CEC cone population and the sharp increase in S-opsin expression in these CEC cells were driven by a shared upstream signaling input. If these two processes respond to the same upstream input, we would expect that they should be coupled and be linked by D-V position. If, however, they do not respond to the same input we would expect that their transitions should be independent. In our model, they are coupled through the T3 gradient and we wanted to test if the experimental retinas were also coupled. As the experimental data show large retina-to-retina variability, we performed 100 additional simulations with parameters sampled from normal distributions parameterized using the experimental variance and checked for overlap of the corresponding probability distributions for two observables.

First, we calculated the probability of having a given CEC population fraction at the S-opsin transition point from both our simulations and experimental retinas (**Fig 9A**). Second, we

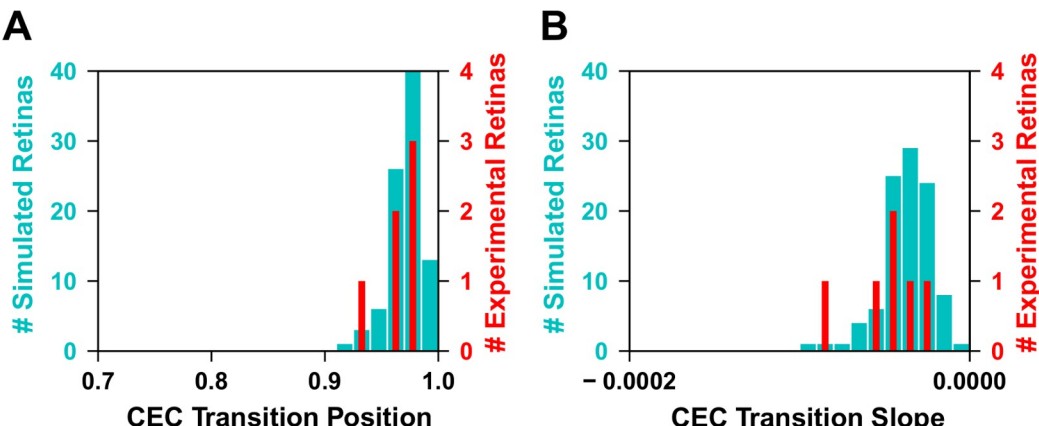

**Fig 9. Correlation between CEC fate and S-opsin transitions. (A)** The fraction of CEC cells at the point where the S-opsin transition is at its midpoint. Data are shown for both experimental (red) and modeled (cyan) retinas. **(B)** The slope of the CEC transition at the S-opsin midpoint, for both experimental (red) and modeled (cyan) retinas. Note: the distributions of only 5 of the 6 retinas are included here, as one of the images had major disruptions at the transition zone due to disecting and mounting.

calculated the rate at which the CEC population decreases at the transition point (**Fig 9B**). As can be seen from both plots, the experimental and simulated probability distributions are in good agreement. In particular, the widths of the experimental probability distributions are similar to the widths from the probabilistic simulations. With the small number of experimental data points, we do not assign a level of statistical significance to the overlaps, but they provide qualitative evidence that the two transitions are in fact coupled through a shared upstream input.

## Comparison of Thrβ2 mutant retinas to wild-type retinas

Finally, to elucidate the effect of the T3 gradient on cone cell patterning in the mouse retina, we dissected and imaged Thrβ2 knockout mutant retinas (ΔThrβ2). In the absence of functional Thrβ2, no M-opsin is expressed [5, 8, 18]. Consistent with previous work, we observed no M-opsin expression in these mutant retinas. Fluorescence from anti-M-opsin antibodies was nonspecific and stained cell and background with equal intensity (**S12 Fig**).

The expression of S-opsin in cones was also markedly different between ΔThrβ2 and WT retinas. Both the density of S-opsin-expressing cones and the expression distribution are flat with respect to the D-V axis (**Fig 10B, S13 Fig**). Also, the relative intensity of the S-opsin signal across the retina was much lower in ΔThrβ2 retinas than the maximum value seen in WT retinas (e.g., from cones in the ventral region). ΔThrβ2 retinas and wildtype retinas were taken at the same time and stained with the same batch of antibody, then imaged with the same laser intensity for comparison of opsin levels. We found that the relative intensity of opsin staining in cones was most similar to the middle region of WT retinas. This effect is consistent with our

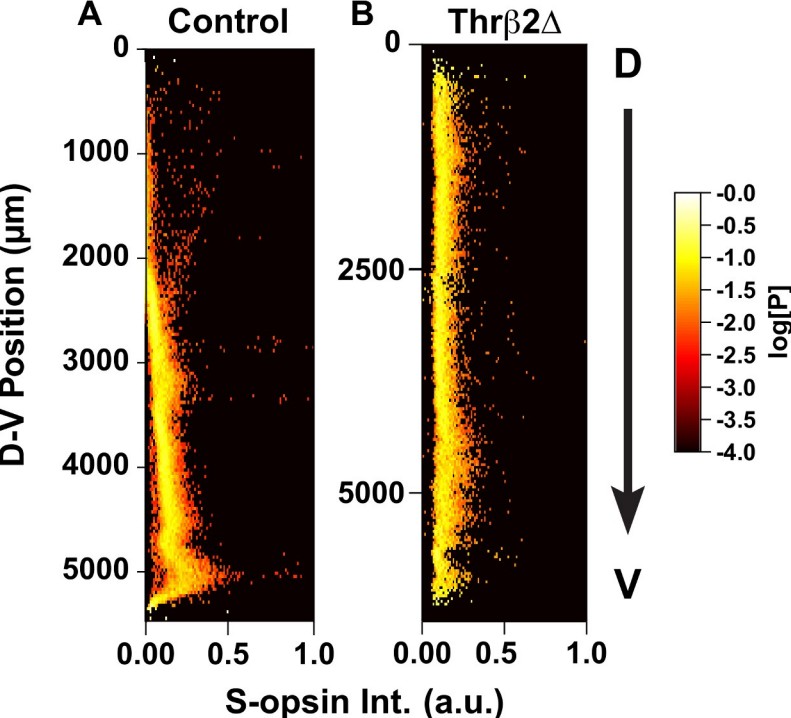

**Fig 10. ThrB2Δ mouse Intensity Plots.** Relative intensity of S-opsin cone cells (X-axis) displayed as a function of dorsal to ventral position. Each point is colored according to the $\log_{10}$[Probability] of expression levels. **(A)** Control Retina, as seen in **Fig 5F**. **(B)** Thrβ2Δ retina.

model in which S-opsin expression is controlled through a combination of negative regulation by active Thrβ2* and positive regulation by inactive Thrβ2 (**Fig 6B4**).

## Discussion

In these studies, we described the distribution of cone photoreceptors in the mouse retina and developed a quantitative model for the specification of binary and graded cell fates in response to D-V regulatory inputs. By using high-resolution microscopy combined with automated image analysis, we expanded on previous studies and mapped the cell fate decisions of cone cells across an entire dorsal to ventral region of the mouse retina. By analyzing cell fates in the context of their position in the tissue, we found that cones could be classified into two sub-classes with a graded gene expression profile changing nonlinearly. This study exemplifies the benefits of quantitatively analyzing populations of cells in a tissue when classifying fate decisions.

In the mouse retina, we defined two cone subtypes, S-only cones and CEC cones, based strictly on opsin expression profiles. Interestingly, the population of S-only cones in the dorsal region have higher S-opsin expression than most S-only cones in the ventral region (**Fig 5J**). These highly expressing S-only cones are found at a steady density across the D-V axis of the retina. It is possible that this subset of evenly distributed, high S-opsin expressing cells could comprise the "genuine" S-cones that connect to blue-cone bipolar neurons [7]. We were not able to mathematically distinguish genuine S-cones from the total population of S-only cones. Together, opsin expression and connectivity suggest three possible distinct cone subtypes: 1. CEC cones that do not connect to blue-cone bipolars, 2. "genuine" S-only cones that connect to blue-cone bipolars, and 3. S-only cones that do not connect to blue-cone bipolars. Examination of connectivity in conditions that perturb thyroid hormone signaling may inform the relationship between opsin expression and connectivity and their relationship to cone subtype.

Our study models how the terminal pattern of opsin expression and cone subtypes can be regulated by thyroid hormone signaling. A next step is to address the temporal dynamics of this process during development. Thyroid hormone signaling through Thrβ2 is necessary and sufficient to induce M-opsin and inhibit S-opsin [8, 18]. S-opsin mRNA is expressed highly in the ventral region and M-opsin mRNA is expressed highly in the dorsal region at P0 [47]. Interestingly, distinct differences in T3 levels conducted on dorsal and ventral halves are only observed by P10 [8]. These observations suggest two main possibilities. First, earlier differences in T3 levels may be cell-type-specific and/or below the threshold for detectability and these subtle differences in TH establish the cone subtype pattern. Second, a two-step mechanism controls patterning whereby opsin expression is (1) initially patterned by a TH signaling independent pathway and then (2) maintained and/or reinforced by TH signaling to determine the terminal pattern. As TH signaling is necessary and sufficient to determine the terminal pattern, our model is consistent with either of these possibilities.

Additional developmental studies support our model for cone fate specification. Daniele et al. 2011 studied the effects when S-opsin was knocked out. Mice lacking S-opsin have a significant number of cone cells in the ventral region that do not express any opsin and have disrupted cone morphology. We hypothesize that this degrading cone cell population is the S-only cones. In the cells of the mid and dorsal retinal regions, the M-opsin protein levels are higher than in wild type mice, presumably due to less competition for translation machinery, and therefore higher translation of M-opsin mRNA transcripts. In addition, other factors that modulate opsin levels in could be added to the model, such as RNA transcript availability [48].

We developed a mathematical model that described both the binary fate specification process of cones and the graded expression of opsins, all driven by an external gradient.

Probabilistic modeling of this complex process generated probability distributions that we used to compare with experimentally observed cell distributions to test hypotheses about the connections between cell fates. Probabilistic modeling is now sufficiently mature to perform detailed simulations of tissue-level cell-fate decisions. Combining probabilistic models with high-throughput microscopy is a powerful tool for helping to understand complex relationships in tissues.

These methods advance our understanding of how regulatory inputs influence complex cellular decisions to specify binary and graded cell fates within the same cell type in the same tissue. A next step will be to integrate more signaling inputs into our model for retinal development. Numerous signaling molecules are expressed in D-V gradients and are involved in retinal and cone cell development. Specifically, retinoic acid is an important morphogen that is expressed at high levels ventrally during development, and then at moderate levels in the dorsal region in the adult mouse [49, 50]. Moreover, further studies would include integrating transcription factor binding partners of Thrβ2 into the model, as Thrβ2 acts as a homodimer and as a heterodimer with RXRγ [8]. This work represents an important first step towards modeling the complex network of interactions that guide binary and graded cell fate specification.

The retina provides an excellent paradigm to study how signaling inputs generate patterns in two dimensions. The next challenge will be developing models for patterning in more complex 3-dimensional neural tissue found in brain structures. Quantitative modeling has enormous potential to integrate multiple signals across a tissue and build networks to better understand and predict the outcomes of development when variables are changed, for instance in disease states.

## Materials and methods

### Ethics statement

All animals were handled in accordance with guidelines of the Animal Care and Use Committees of Johns Hopkins University under the protocl number MO19A219. All efforts were made to minimize the the stress to animals and the number of animals used.

### Animals

Mice (strain C57BL/6) were housed under a 12 h light:12 h dark (T24) cycle at a temperature of 22˚C with food and water *ad libitum*. Male and female mice were separately housed in plastic translucent cages with steel-lined lids in an open room. Ambient room temperature and humidity were monitored daily and tightly controlled. Wild-type mice (C57BL/6; Jackson Laboratory), and *Thrb^{tm2Df}* mutant mice (gift from the Forrest Lab) were used in this study. *Thrb^{tm2Df}* mutant mice specifically knock out expression of Thrβ2, and leave Thrβ1 intact as previously described [18]. Retinal dissections were performed on mice at 2–8 months old.

### Immunohistochemistry

Retinas were dissected in PBS, then fixed in fresh 4% formaldehyde in PBS for 1 hour. The dorsal portion of the retina was marked with a cut. Tissue was rinsed 3X for 15 min in PBS. Retinas were incubated for 2 hours in blocking solution (0.2–0.3% Trition X-100, 2–4% donkey serum in PBS). Retinas were incubated with primary antibodies in blocking solution overnight at 4˚C. Retinas were washed 3X for 30 min in PBS, and then incubated with secondary antibodies in blocking solution for 2 hours at room temperature. At the end of staining, retinas

were cut to lay flat on a slide, and were mounted for imaging in slow fade (S36940, Thermo Fisher Scientific).

## Antibodies

Primary antibodies were used at the following dilutions: polyclonal goat anti-SW-opsin (1:200) (Santa Cruz Biotechnology), polyclonal rabbit anti-LW/MW-opsins (1:200) (Millipore). All secondary antibodies were Alexa Fluor-conjugated (1:400) and made in donkey (Molecular Probes).

## Microscopy and image processing

Fluorescent images were acquired with a Zeiss LSM780 or LSM800 laser scanning confocal microscope. Confocal microscopy was performed with similar settings for laser power, photo-multiplier gain and offset, and pinhole diameter. Whole retinas were imaged with a 10X objective, and maximum intensity projections of z-stacks (5–80 optical sections, 4.9 μm step size) were rendered to display all cones imaged in a single retina. Retinal strips were imaged with a 20X objective, and maximum intensity projections of z-stacks (5–80 optical sections, 1.10 μm step size) were rendered to display all cones imaged in a single retina. ΔThrβ2 retinas and wild-type retinas were taken at the same time and stained with the same batch of antibody, then imaged with the same laser intensity for comparison of opsin levels.

## Segmentation of cone cells from microscopy images

Microscopy images were analyzed using a custom parallel image processing pipeline in Biospark [51]. Briefly, each fluorescence channel was first normalized and filtered to remove small bright features. Then, each remaining peak in fluorescence intensity was identified and an independent active contour segmentation [52] was performed starting from the peak. If the resulting contour passed validation checks it was included in the list of segmented cone cells for the channel. Finally, the outer segment boundaries of cone cells were reconciled across both channels to obtain a complete list of identified cells. Full details are given in the SI Methods.

## Modeling cone cells fate decisions and opsin expression

Multiscale modeling of the retina strip was performed using a hybrid deterministic-probabilistic method. Diffusion of T3 in the microenvironment of the retinal strip was modeled using the diffusion partial differential equation (PDE). The PDE was solved using an explicit finite difference method. The cone cells were modeled using the chemical master equation (CME) to describe the probabilistic reaction scheme implementing the cell fate decision-making. The CME for each of the 23,760 cone cells was independently sampled using Gillespie's probabilistic simulation algorithm [53]. Reconciliation between the CME trajectories and the PDE microenvironment was done using a time-stepping approach. Complete mathematical details of the model and simulation methods are available in the SI Methods. All simulations were performed using a custom solver added to the LMES software [54], which is available on our website: https://www.robertslabjhu.info/home/software/lmes/.

## Supporting information

**S1 Text. Supplementary methods.**
(PDF)

**S1 Table. Retina image names and genotypes.**
(PDF)

**S2 Table. The number of cells identified per subtype by hand (H) and computer (C) annotation.**
(PDF)

**S3 Table. Parameters used in the microenvironment model.**
(PDF)

**S1 Fig. Outer segment size by expression class.** The probability distribution for the outer segment area is shown for each expression class. Columns shows different expression classes, as labeled. Rows show different retinas (RXX). No systematic differences were observed between the classes, but note that retina-to-retina variability in outer segment area is present due to variations in mounting and subsequently the angle of image capture for cone cell outer segments.
(PDF)

**S2 Fig. Fitting of cell expression data.** Fraction of cells expressing (left) M-opsin and (right) S-opsin by position along the D-V axis. The data from the microscopy analysis (x) are overlaid with the best fit (line) to a fitting function (see text). Rows show different retinas (RXX).
(PDF)

**S3 Fig. Comparison of D-V profiles between retinas.** Overlap of the fraction of cells expressing (left) M-opsin and (right) S-opsin aligned to the transition midpoint as determined from the S-opsin expression profile.
(PDF)

**S4 Fig. S-only cell fraction.** Fraction of cells expressing only S-opsin by position along the D-V axis. The data from the microscopy analysis (x) are overlaid with the best fit (line) to a fitting function (see text). Rows show different retinas (RXX).
(PDF)

**S5 Fig. Correlation between S- and M-opsin in retinal cells.** Joint probability distributions for the abundance of S-opsin (blue intensity) and M-opsin (green intensity) in cells. Rows show different retinas (RXX). Colors range from $\log_{10}[P] = -2$ (white/yellow) to $\log_{10}[P] = -6$ (red/black).
(PDF)

**S6 Fig. Correlation between S- and M-opsin in retinal cells.** Joint probability distributions for the abundance of S-opsin (blue intensity) and M-opsin (green intensity) in cells. Columns show cells binned from four different regions according to distance from the transition midpoint. Rows show different retinas (RXX). Colors range from $\log_{10}[P] = -2$ (white/yellow) to $\log_{10}[P] = -4$ (red/black).
(PDF)

**S7 Fig. Expression of S- and M-opsin in retinal cells.** Probability distribution for the abundance of (left) M-opsin and (right) S-opsin in cells by distance from the transition midpoint. Rows show different retinas (RXX). Colors range from $\log_{10}[P] = 0$ (white/yellow) to $\log_{10}[P] = -4$ (red/black).
(PDF)

**S8 Fig. Fitting of cell expression intensity data.** Mean intensity in all cells of (left) M-opsin and (right) S-opsin by position along the D-V axis. The data from the microscopy analysis (x) are overlaid with the best fit (line) to a fitting function (see text). Rows show different retinas

(RXX).
(PDF)

**S9 Fig. Expression in modeled cell populations.** Mean fraction of cells in various cell populations along the D-V axis from numerical simulations of the model. Plots show the mean value computed from 100 independent simulations.
(PDF)

**S10 Fig. Opsin concentrations in modeled cells.** Probability distribution of the abundance of S-opsin (blue intensity) and M-opsin (green intensity) in cells along the D-V axis from numerical simulations of the model. Distributions were computed from 100 independent simulations.
(PDF)

**S11 Fig. Correlation between S- and M-opsin in modeled cells.** Joint probability distributions for the abundance of S-opsin (blue intensity) and M-opsin (green intensity) in cells located in ∼250μm wide bins along the D-V axis. Colors range from log_10[P] = −2 (white/yellow) to log_10[P] = −5 (red/black). Distributions were computed from 100 independent simulations. The low density tails leading to 0,0 are from cells that were sampled during the process of switching phenotypes.
(PDF)

**S12 Fig. Analysis of pixel intensities in images of ΔTHRβ2 cells.** (left) Joint probability distribution of the blue and green intensity of pixels located either inside of cell boundaries (RXX CELL) or the background outside of cells (RXX BG) as indicated. Colors range from log_10[P] = 0 (white/yellow) to log_10[P] = −8 (red/black). (center) Probability for a pixel of the indicated type to have a particular blue intensity (solid line) compared with the distribution for all pixels (dashed line). (right) The same for green intensity. ΔTHRβ2 cells do not exhibit green expression above background.
(PDF)

**S13 Fig. Expression of S-opsin in ΔTHRβ2 retinal cells.** Probability distribution for the abundance of S-opsin in cells by distance along the D-V axis. Rows show different ΔTHRβ2 retinas (RXX). Colors range from log_10[P] = 0 (white/yellow) to log_10[P] = −4 (red/black).
(PDF)

**S14 Fig. Mean retina description.** Comparison of the fits for individual retinas (dashed lines) with our hypothetical mean retina used for model parameterization (solid line) along the D-V axis. The top row shows a comparison of the fraction of cells expressing M- and S- opsin, respectively. The middle row shows the fraction of FD(S) cells. The bottom row shows the mean M- and S-opsin expression intensity, respectively.
(PDF)

**S15 Fig. Best fit parameterization.** Comparison of the best fit model parameterization (blue) with the hypothetical mean retina (black). The top row shows a comparison of the fraction of cells expressing M- and S- opsin, respectively. The middle row shows the fraction of FD(S) cells. The bottom row shows the mean M- and S-opsin concentration per cell, respectively.
(PDF)

## Acknowledgments

The authors acknowledge the contribution of Haiqing Zhao of Johns Hopkins University for providing animals for this study and training for K.C.E., as well as Doug Forrest of the NIH for providing ΔThrβ2 mice.

## Author Contributions

**Conceptualization:** Robert J. Johnston, Jr, Elijah Roberts.

**Data curation:** Elijah Roberts.

**Formal analysis:** Elijah Roberts.

**Funding acquisition:** Robert J. Johnston, Jr, Elijah Roberts.

**Investigation:** Kiara C. Eldred, Cameron Avelis, Elijah Roberts.

**Methodology:** Kiara C. Eldred, Elijah Roberts.

**Project administration:** Kiara C. Eldred, Robert J. Johnston, Jr, Elijah Roberts.

**Resources:** Elijah Roberts.

**Software:** Elijah Roberts.

**Supervision:** Robert J. Johnston, Jr, Elijah Roberts.

**Validation:** Robert J. Johnston, Jr, Elijah Roberts.

**Visualization:** Elijah Roberts.

**Writing – original draft:** Kiara C. Eldred.

**Writing – review & editing:** Kiara C. Eldred, Robert J. Johnston, Jr, Elijah Roberts.

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
