## [Decision Letter · Decision Letter 0]

7 Nov 2019

Dear Dr Roberts,

Thank you very much for submitting your manuscript 'Modeling binary and graded cone cell fate patterning in the mouse retina' for review by PLOS Computational Biology. Your manuscript has been fully evaluated by the PLOS Computational Biology editorial team and in this case also by independent peer reviewers. The reviewers appreciated the attention to an important problem, but raised some substantial concerns about the manuscript as it currently stands. While your manuscript cannot be accepted in its present form, we are willing to consider a revised version in which the issues raised by the reviewers have been adequately addressed. We cannot, of course, promise publication at that time.

Sincerely,

Pedro Mendes, PhD

Associate Editor

PLOS Computational Biology

Mark Alber

Deputy Editor

PLOS Computational Biology

[LINK]

Reviewer's Responses to Questions

**Comments to the Authors:**

Reviewer #1: In this manuscript, Eldred et al. analyze the distribution of M- and S-opsin expression in the mouse retina and use an image analysis approach to model the choice of cone cells to express M-opsin, S-opsin, or both. They first characterize in detail the distribution of the two opsins in six wild-type mouse retinas and reproduce what was already known, i.e. that M-opsin is enriched in the dorsal part of the retina, while S-opsin the ventral part. Reasonably, co-expressing cones are mostly enriched in the middle and mid-ventral part. They then argue that their data agree with a previous study (Haverkamp et al 2005), that stated that instead of having three different populations of cone cells (S-expressig, M-expressing, and co-expressing, there exist two population (S-expressing and co-expression competent cones) and pursue to model these two decisions a cone cells has to make: a) a binary decision to express S-opsin or be competent to express both, and b) a graded decision to express different levels of the two opsins, based on their position on the D-V axis. Interestingly, both decisions are influenced by the presence of thyroid hormone T3, whose levels are higher in the dorsal part of the retina and decrease along the D-V axis. Then, they try to show that their model can recapitulate their wild-type data, as well as the reduction of expression of S-opsin in Thrb2 mutant retinas.

While the paper is well done, well written, and interesting to read and the question is definitely very intriguing, I don’t believe that it offers any biological, computational, or conceptual advance to warrant publication in PLoS Computational Biology. The authors repeat what was already known in a quantitative manner and come up with a not-so accurate model, which was already proposed in the past. Moreover, the data that support that their model recapitulates the observed data (i.e. Figures 4, 8, and 9) are the least convincing figures of the manuscript. Please see the major issues with these figures:

- In Fig. 4, while I have no reason to disagree that there are two populations of cone cells (S-expressing and co-expression competent), this is not shown or supported by these data. The authors manually draw a white dashed line that agrees with their hypothesis, but they could have as easily drawn another white line between the co-expressing population and the M-expressing population, where there is a dip in the yellow portion of the heatmap (approximately in position (0.15,0.02)). I would be far more comfortable with these data, if the white-dashed line was drawn in a “computational” way, rather than manually.

- In Fig. 8, while the authors argue that their model recapitulates their observations, in fact now the difference between S-opsin expressing cells and M-opsin expressing cells is now more evident. I would have loved to seen the equivalent of Figure 4A in Figure 8, as well. I believe that if the authors overlay Figures F-J, there will be no “dip” between M-opsin expressing cells and co-expressing cells (i.e. the yellow portion will be continuous). This would contradict the observed data and will also justify my previous comment about how the white dashed line was drawn. This is also illustrated by the fact that in Figures 8F-J, the yellow (“hot”) region doesn’t even “approach” the white dashed line, while in Figures 8D-E, it almost overlaps with it. Again, I am not disagreeing with the biology, rather with the interpretation that the model recapitulates observed data.

- This becomes even stronger in Figure 9, where the authors argue that experimental and simulated data agree with each other, while this is definitely not is shown in this figure.

Reviewer #2: General comments:

The manuscript by Eldred et al. offers fresh views and approaches to understanding cell fate decisions, using a highly accessible system within the mouse retina: patterns of expression of M vs. S (vs. both) opsins. While the roles of TRb2, and gradients of T3 for patterning M vs. S expression have been known for some time, the current study and analysis provides finer detail and an interesting new model – advancing the concept of a binary and then graded cell fate. This study appears to be appropriate for PLOS Computational Biology, and is mostly written in a manner that is understandable by a broad audience (not strictly a computational audience). However, some key methods information is lacking, there is room for clarification in several places, and the manuscript misses an opportunity to compare with existing developmental literature.

Specific comments:

Abstract. Within the abstract, the authors use the term “stochastic model,” and while this refers to their tool for estimating probability distributions, it is also confusing because there are a number of “stochastic models” for cone fate decisions out there in the retina world, and these models mean something very different from the binary + graded paradigm the authors propose. This reviewer suggests simply striking out the word “stochastic” from the abstract to avoid this confusion.

Page 3, 2nd paragraph. Superimposed upon the perfect grid of the zebrafish cone mosaic is variable patterning of opsin expression within the L and M populations.

Page 3, 3rd paragraph. Upon reading this section, this reviewer expected the manuscript to contain developmental studies to back up the predicted two-step cone patterning process. The authors should be clear here that the model is based upon analysis of adult retina, and is predictive of development.

Page 3, bottom paragraph. Here and in the Methods, the authors should state the age(s) of animals that were sampled. This information is not clear in the Methods, and readers need to know this at the outset of the Results.

Page 6, top. Was the deterministic diffusion equation informed at all by actual data, or at least consistent with what is known? Roberts et al. (2006) reports dorsal and ventral T3 levels from retinas collected at P0, P4, and P11.

Page 6, middle (and Fig. 6). The authors use the term “stochastically” again in a somewhat confusing/misleading manner, particularly because the “stochastic” progression is influenced by a diffusible ligand binding to its receptor. The component of the model shown in Fig. 6B(2) does not look stochastic.

Page 7, middle. The authors refer to their stochastic simulations, and again the word is unfortunately misleading (even if accurately used). Perhaps here, or earlier in Results, the meaning of stochastic modeling/simulations could be defined?

Page 9. The authors should compare their predictions with what is known of the developmental time-course of M and S opsin patterning (and formation of the T3 gradient) within the literature. Do the two-step results bear out experimentally, or are they at least consistent with the reports from Roberts et al. (2006) and Aavani et al. (2017), among others? If not, how could any conflicts be resolved by either modifying the model or performing experiments? Also missing from the discussion is a mention of the interesting outcome of knocking out S opsin (Daniele et al., 2011) – M opsin expression increases, speculatively due to reduced competition for translational machinery. This outcome suggests factors other than those offered in the current manuscript can regulate opsin expression levels.

Page 16. Probably the antibodies were polyclonals, but this should be stated.

Page 16. Do the authors mean outer segment boundaries rather than cell boundaries?

Figure 9 took a while to appreciate, and may be easier for a reader to grasp if the X axes had slightly greater scales. As presented, the first impression is that the experimental and simulated positions and slopes do not map on to each other very well.

Reviewer #3: The manuscript titled "Modeling binary and graded cone cell fate patterning in the mouse retina" presents image analysis and computational modeling results that show how photoreceptor cells in the mouse retina develop and differentiate.

This paper is extremely well written, their conclusions are well support by their data, and all of their material is publicly available for reproduction of their results. I see no reason that this paper should not be published immediately.

**Have all data underlying the figures and results presented in the manuscript been provided?**

Reviewer #1: Yes

Reviewer #2: Yes

Reviewer #3: Yes

PLOS authors have the option to publish the peer review history of their article (what does this mean?). If published, this will include your full peer review and any attached files.

Reviewer #1: No

Reviewer #2: No

Reviewer #3: Yes: Brian Drawert

---

## [Decision Letter · Decision Letter 1]

16 Jan 2020

Dear Dr. Roberts,

Thank you very much for submitting your manuscript "Modeling binary and graded cone cell fate patterning in the mouse retina" for consideration at PLOS Computational Biology. As with all papers reviewed by the journal, your manuscript was reviewed by members of the editorial board and by several independent reviewers. The reviewers appreciated the attention to an important topic. Based on the reviews, we are likely to accept this manuscript for publication, providing that you modify the manuscript according to the review recommendations.

Could you please supply the data on the "no morphological changes" result, as indicated by Reviewer #2.

Sincerely,

Pedro Mendes, PhD

Associate Editor

PLOS Computational Biology

Mark Alber

Deputy Editor

PLOS Computational Biology

[LINK]

Reviewer's Responses to Questions

**Comments to the Authors:**

Reviewer #1: The authors responded satisfactorily to my comments and, indeed, the presentation of the data is much clearer now. While I stand by my opinion that there is no biological, computational, or conceptual advance, it is true that this level of quantitative analysis is not common in developmental biology papers.

Reviewer #2: The manuscript has improved, thanks to the authors for their careful attention to responding to all reviewer comments and suggestions.

Only one comment/concern: The removal of (data not shown) does not seem the best way to respect the journal's requirements. The outcome stated on page 6, that no morphological or size differences were identified, can certainly be shown as a supplemental figure, or added to an existing supplemental figure.

**Have all data underlying the figures and results presented in the manuscript been provided?**

Reviewer #1: Yes

Reviewer #2: Yes

PLOS authors have the option to publish the peer review history of their article (what does this mean?). If published, this will include your full peer review and any attached files.

Reviewer #1: No

Reviewer #2: No
---

## [Editor Report · Decision Letter 2]

27 Jan 2020

Dear Dr. Roberts,

We are pleased to inform you that your manuscript 'Modeling binary and graded cone cell fate patterning in the mouse retina' has been provisionally accepted for publication in PLOS Computational Biology.

Before your manuscript can be formally accepted you will need to complete some formatting changes, which you will receive in a follow up email. A member of our team will be in touch within two working days with a set of requests.

Best regards,

Pedro Mendes, PhD

Associate Editor

PLOS Computational Biology

Mark Alber

Deputy Editor

PLOS Computational Biology

---

## [Editor Report · Acceptance letter]

2 Mar 2020

PCOMPBIOL-D-19-01633R2 

Modeling binary and graded cone cell fate patterning in the mouse retina

Dear Dr Roberts,

I am pleased to inform you that your manuscript has been formally accepted for publication in PLOS Computational Biology. Your manuscript is now with our production department and you will be notified of the publication date in due course.

With kind regards,

Matt Lyles
